# Disordered Protein Tail Is Wagging Poly(ADP-ribosyl)ation

**DOI:** 10.3390/ijms26178166

**Published:** 2025-08-22

**Authors:** Guillaume Bordet, Yaroslava Karpova, Saraynia Espeseth, Gavin Mitzel, Zachary Bigelow, Alexei V. Tulin

**Affiliations:** Department of Biomedical Sciences, School of Medicine and Health Sciences, University of North Dakota, 501 North Columbia Road, Stop 9061, Grand Forks, ND 58202, USA; guillaume.bordet@und.edu (G.B.); iaroslava.karpova@und.edu (Y.K.); saraynia.espeseth@und.edu (S.E.); gavin.mitzel@und.edu (G.M.); zachary.bigelow@und.edu (Z.B.)

**Keywords:** PARG, PARP-1, pADPr, lifespan, regulation of localization, intrinsically disordered regions

## Abstract

Intrinsically disordered regions (IDRs) are present in nearly all proteins, often accounting for more than 40% of their amino acid sequence. Unlike structured domains, IDRs lack sequence or structural conservation across species while maintaining conserved biological functions. Here, we discovered that the previously uncharacterized disordered tail region of Poly(ADP-ribose) glycohydrolase (PARG) controls its localization and activity. Despite its structural divergence, this domain supports conserved regulatory functions across species. Deletion of the disordered tail results in cytoplasmic mislocalization, aberrant accumulation in the nucleolus, impaired chromatin association, and reduced enzymatic activity. Mass spectrometry analysis reveals that this disordered region mediates interactions with nuclear transport factors, post-translational modification enzymes, and chromatin-associated complexes. Together, these results demonstrate that the disordered tail region of PARG acts as a regulatory hub that integrates multiple layers of control to ensure proper subcellular localization and chromatin function.

## 1. Introduction

Proteins were traditionally thought to adopt well-defined, three-dimensional structures that determine their function. However, it is now recognized that nearly all proteins contain intrinsically disordered regions (IDRs), which often comprise more than 40% of their amino acid sequence [1]. In some cases, particularly in chromatin-associated proteins such as chromatin remodelers and transcription factors, up to 80% of the sequence can be disordered [2]. These flexible regions confer structural plasticity, enabling proteins to engage in diverse and transient interactions essential for dynamic cellular processes. Despite their prevalence, the mechanistic contributions of IDRs to protein function remain poorly understood.

The Poly(ADP-ribose) (PAR) pathway is a fundamental regulatory system that governs chromatin dynamics, and it is essential for cellular differentiation, development, and malignancy control [3,4,5,6,7,8,9]. This pathway is driven by the opposing activities of Poly(ADP-ribose) polymerase 1 (PARP1) and Poly(ADP-ribose) glycohydrolase (PARG). PARP1 modifies chromatin structure by synthesizing poly(ADP-ribose) (pADPr) chains on chromatin-associated proteins, including itself, leading to chromatin loosening and allowing the transcription machinery to access and bind DNA [10,11]. Conversely, PARG degrades pADPr, restoring chromatin compaction associated with gene silencing [7,12]. Although both enzymes interact with chromatin, their distinct distribution patterns suggest that PARG plays a unique regulatory role beyond simply counteracting PARP1, indicating additional mechanisms of recruitment and function [7].

Over the past decade, research has largely focused on PARP1, while PARG has remained understudied, leaving fundamental questions about its regulation unanswered. Recent findings, including work from our lab, have demonstrated that PARG plays an active role in transcriptional regulation beyond that of passive PARP1 antagonist. Loss of *parg* function results in developmental arrest in both mammals and Drosophila, underscoring its critical role in gene regulation [6,12,13].

In this study, we examine a previously unrecognized disordered tail region of PARG that appears to be central to its regulation. We show that this domain governs a remarkable spectrum of regulatory activities, including nuclear-cytoplasmic trafficking, post-translational modifications, chromatin recruitment, and protein stability, ensuring proper PARG activity and localization.

## 2. Results

### 2.1. Despite Structural Differences, Human PARG Is Fully Functional in Drosophila

Human PARG (hPARG) is a 976-amino acid protein composed of an N-terminal domain (residues 1–360) and a catalytic domain (residues 361–976) [6,14] (Figure 1A). In contrast, *Drosophila melanogaster* PARG (dPARG) is a 768-amino acid protein with its catalytic domain located at the N-terminal (residues 1–607), which includes key catalytic residues (E385, E386, and E394), followed by a C-terminal domain (residues 608–768) of unknown function [6] (Figure 1A). While the catalytic domains of human and Drosophila PARG are structurally highly conserved (Figure 1B), their non-catalytic regions are strikingly different in that the human N-terminal and Drosophila C-terminal regions are intrinsically disordered regions (IDRs) with no obvious structural similarity [15,16,17] (Figure 1A,B, Appendix A). Disorder prediction analyses indicate that such disordered tails are present in PARG orthologs across species (Appendix A), and sequence alignment reveals that these regions are highly divergent, with no detectable conservation (Appendix A).

IDRs have recently emerged as key regulators of protein function through diverse mechanisms, including protein–protein interactions [18]. In particular, IDRs in PARG protein have no stable 3D structure, but rather exist as a collection of interconverting conformations destined to perform a variety of cellular processes. We hypothesized that these highly flexible and functional PARG disordered tails controls PARG activity. To test this, we generated transgenic Drosophila lines expressing GFP-tagged versions of either human or Drosophila PARG (hereinafter termed as hPARG and PARG-WT, respectively) (Figure 1C). Remarkably, both hPARG and PARG-WT fully rescued the developmental arrest observed in *parg* null mutants and supported a normal lifespan (Figure 1D). Moreover, both proteins maintained low levels of pADPr and exhibited identical subcellular localization patterns across all tested tissues (Figure 2 and Appendix A). Therefore, despite their profound structural differences, these results demonstrate that hPARG is fully functional and properly regulated in Drosophila, underscoring a deeply conserved mechanism of PARG function across species.

### 2.2. The Disordered Tail Controls PARG Activity and Subcellular Localization

To determine the functional significance of the Drosophila PARG disordered tail, we generated transgenic lines expressing a GFP-tagged version of the catalytic domain alone (residues 1–607, termed as PARG-Cat) (Figure 1C). While PARG-Cat -expressing flies were viable in a *parg* null mutant background, they exhibited severe phenotypic defects, including 50% developmental delay before pupation and 30% reduction in lifespan compared to PARG-WT (Figure 1D and Appendix A). Additionally, PARG-Cat-expressing flies displayed significantly elevated pADPr levels, similar to *parg* null mutant, indicating impaired PARG activity (Figure 2A,B and Appendix A). Beyond its functional deficits, PARG-Cat exhibited severely disrupted subcellular localization. The major fraction of PARG-Cat mislocalized to the cytoplasm, while the nuclear fraction aberrantly accumulated in the nucleolus, an organelle where PARG is not normally detected under physiological conditions (Figure 2C,D and Appendix A). Moreover, Western blot analysis of PARG-Cat revealed two distinct bands with the lower band corresponding to the expected molecular weight (Appendix A). This suggests that PARG-Cat undergoes post-translational modifications. Fractionation experiments further demonstrated that only the higher molecular weight form is present in the nucleus, indicating that this modification may be required for nuclear retention (Appendix A). Altogether, these data reveal PARG disordered tail controls the proper subcellular distribution and functional integrity, preventing both cytoplasmic mislocalization and aberrant nucleolar accumulation.

### 2.3. The Disordered Tail Is Sufficient to Control PARG Subcellular Localization

To determine whether the Drosophila PARG disordered tail is sufficient to direct proper subcellular localization, we generated transgenic flies expressing only the C-terminal tail region of PARG (residues 608–768) fused to GFP (termed as PARG-D-tail) (Figure 1C). As expected, PARG-D-tail, which lacks catalytic activity, failed to rescue the developmental arrest of *parg* mutants and exhibited elevated pADPr levels comparable to those observed in *parg* mutants (Figure 2A,B and Appendix A). Surprisingly, despite its lack of enzymatic function, PARG-D-tail did exhibit a subcellular localization pattern identical to that of PARG-WT across all tested tissues (Figure 2C,D and Appendix A). These findings demonstrate that the intrinsically disordered tail region of PARG is sufficient to dictate proper subcellular localization, reinforcing its critical role as a regulatory element in PARG function.

### 2.4. The Disordered Tail Controls PARG Nuclear–Cytoplasmic Trafficking and Ensures Functional Nuclear Localization

It is proposed that IDRs facilitate protein–protein interactions through multiple weak interactions distributed across the domain, rather than through a single high-affinity binding site, which would be characteristic of a loose collection of dynamic interconverting conformations, not just linkers [18]. Accordingly, we hypothesized that the PARG disordered tail controls the subcellular localization by mediating interactions with nuclear transport machinery.

To test this, we performed immunoprecipitation followed by mass spectrometry (IP-MS) on transgenic flies expressing GFP-tagged PARG-WT, PARG-Cat or PARG-D-tail. A total of 387 proteins were identified as significantly interacting with PARG (Appendix A). Of these, 289 proteins interacted with the catalytic domain (Appendix A) and 191 with the disordered tail (Appendix A). Gene ontology (GO) analysis revealed distinct functional roles for these interacting proteins.

The most striking consequence of deleting the PARG disordered tail is its severe mislocalization to the cytoplasm, even while a fraction of the protein remains detectable in the nucleus. Conversely, in the absence of the catalytic domain (PARG-D-tail), PARG is retained exclusively in the nucleus, similar to PAG-WT. Proteins larger than ~40–50 kDa require active transport into the nucleus through the Ran/Importin system whereby a small GTPase and transport receptors facilitate the movement of molecules [19]. Therefore, we reasoned that both the catalytic and disordered tail of PARG might interact with nuclear import machinery, whereas only the catalytic domain interacts with nuclear export machinery.

Our mass spectrometry data strongly support this hypothesis. We identified both alpha- and beta-importins interacting with the catalytic and disordered tail regions of PARG (Figure 3A,B). Among them, the alpha-importin Pendulin (Pen) was the most enriched. In contrast, we detected several nuclear export proteins, including Embargoed (Emb, the Drosophila ortholog of CRM1) and its cofactor RanBP3, which only interacted with the catalytic domain (Figure 3A,B). CRM1 is a key mediator of nuclear export, facilitating protein translocation through the nuclear pore complexes in conjunction with RanGTP [20].

These results reveal a crucial role for the PARG disordered tail in modulating the balance between nuclear import and export. In PARG-WT, interactions with importins are strengthened by contributions from both the catalytic and disordered tail regions, ensuring strong nuclear localization. However, in the absence of the disordered tail, the weaker interaction with importins shifts the balance toward export, leading to cytoplasmic accumulation. Conversely, PARG-D-tail remains exclusively nuclear owing to its retained import interactions and lack of interaction with nuclear export machinery.

We next investigated whether nuclear localization could be attributed to the predicted nuclear localization signal (NLS) within the disordered tail [6] (Figure 4A). To test this, we generated a PARG construct lacking this sequence (PARG-Δ685-710) (Figure 4A). PARG-Δ685-710 localized predominantly to the cytoplasm, although a fraction was still detected in the nucleus but was completely absent from the nucleolus (Figure 4B,C). Remarkably, when expressed in a *parg* mutant background, PARG-Δ685-710 maintained low pADPr levels comparable to PARG-WT, indicating that the remaining nuclear fraction is fully functional (Figure 4D,E and Appendix A).

In contrast, forcing nuclear import of the catalytic domain by fusing it to a constitutive NLS (PARG-Cat-cNLS) eliminated cytoplasmic localization but resulted in aberrant nuclear distribution, with strong enrichment in the nucleolus and on chromatin, and minimal presence in the soluble nucleoplasm (Figure 4B,C). PARG-Cat-cNLS accumulated high pADPr levels, indicating loss of functional activity despite its nuclear localization (Figure 4D,E and Appendix A).

Together, these results demonstrate that the disordered tail controls not only the balance between nuclear import and export but also the correct subnuclear targeting, which is critical for the full enzymatic activity of PARG.

### 2.5. Interaction with the Protein Degradation Machinery Is Primarily Mediated Through the PARG Catalytic Domain

Nuclear proteins are often exported to cytoplasm for degradation [21]. Notably, CRM1 has been shown to facilitate the nuclear export of the CDK inhibitor p27 during G1 phase, leading to its cytoplasmic degradation via the ubiquitin-proteasome system (UPS) [22]. Given that PARG interacts with nuclear export machinery, we hypothesized that its cytoplasmic localization could expose it to degradation pathways.

Proteins are targeted for degradation through poly-ubiquitination, a process mediated by E3-ubiquitin ligases that specifically recognize and tag proteins for proteasomal degradation [23]. Our mass spectrometry analysis identified two E3-ubiquitin ligases interacting with PARG: Slmb, a key component of the Skp/Cullin/F-box (SCF) complex, which regulates protein turnover in both the nucleus and cytoplasm [24], and Stub1, a cytoplasmic ubiquitin ligase primarily involved in chaperone-assisted protein degradation and stress-induced protein quality control [25] (Figure 3A,C). In addition to these E3 ligases, we identified two E1 enzymes, Uba1 and Uba5, which are involved in ubiquitin activation and processing.

The preferential interaction of PARG degradation machinery with the catalytic domain suggests a potential mechanism in which nuclear export promotes PARG degradation by exposing it to cytoplasmic E3 ligases, such as Stub1. This reinforces the idea that PARG nuclear-cytoplasmic trafficking is not only crucial for its function but also a major determinant of its stability and turnover. Importantly, Slmb plays a crucial role in the degradation of nuclear and cytoplasmic substrates, and its interaction with PARG suggests that PARG turnover is tightly controlled by degradation pathways, even within the nucleus.

### 2.6. Distinct Kinases Regulate PARG Phosphorylation at Catalytic and Disordered Tail Regions

We previously showed that PARG stability and function are tightly regulated by phosphorylation, influencing key biological processes, such as germline stem cell renewal/differentiation balance and longevity [6]. We identified two major phosphorylation sites in Drosophila PARG: Ph1, located at the beginning of the catalytic domain (Serine 69 and Serine 73), and Ph2, located at the start of the disordered tail (Serine 621, Serine 624, Serine 627, and Serine 628). Our prior data suggested that GSK3 and CKII kinases were the candidates most likely responsible for PARG phosphorylation [6].

To further explore this, we analyzed our mass spectrometry data for kinase interactions. Surprisingly, we found no overlap between the kinases that interact with the catalytic domain (where Ph1 is located) and those that interact with the disordered tail (where Ph2 is located) (Figure 3A,D). This suggests that Ph1 and Ph2 are phosphorylated by distinct kinases. Specifically, the GSK3 kinase Shaggy (Sgg) was the most enriched interactor of the catalytic domain, while CKIIalpha was the most enriched interactor of the disordered tail, the same two kinase families we previously identified as likely regulators of PARG phosphorylation.

We also previously reported that constant phosphorylation of PARG results in a less active protein, suggesting that a dynamic phosphorylation-dephosphorylation cycle is necessary for proper regulation [6]. Following up this reasoning, our mass spectrometry data identified Tws-PP2A as the only phosphatase complex interacting with PARG (Figure 3A,E).

This complex consists of the scaffolding protein Mts, the regulatory subunit Tws, and the catalytic subunit PP2A-29b [26]. Interestingly, PP2A-29b and Mts interact with both the catalytic and disordered tail regions, while Tws interacts exclusively with the catalytic domain. This suggests that the same phosphatase complex dephosphorylates both Ph1 and Ph2 sites, even though their phosphorylation is mediated by different kinases.

Taken together, these data reinforce the idea that PARG phosphorylation is dynamically regulated through distinct kinase-phosphatase interactions at its catalytic and disordered tail regions, likely influencing both its enzymatic activity and stability in response to cellular cues.

### 2.7. SUMOylation as a Potential Mechanism of PARG Regulation

Western blot analysis of PARG-Cat reveals a distinct double-band pattern (Appendix A) with the lower band corresponding to the expected size and the higher band showing a ~10–15 kDa shift, indicative of post-translational modifications (PTMs) with SUMOylation being a strong candidate. SUMOylation is a process whereby a ubiquitin-like modifier protein covalently attaches to lysine residues on target proteins. This process involves three enzymatic steps mediated by E1-activating enzymes, E2-conjugating enzymes, and E3-ligases with E3-ligases conferring specificity by directing the modification of target proteins [27].

Our analysis identified Nup358, a nuclear pore component with E3-ligase activity, as a specific interactor of the PARG catalytic domain (Figure 3A,F) [28]. This suggests that Nup358 may mediate SUMOylation of the catalytic domain, potentially influencing PARG’s nuclear-cytoplasmic trafficking and/or activity. In addition, we found that both subunits of the heterodimeric E1-activating complex, Uba2 and Aos1, which is required for the first step of SUMOylation, are associated with the catalytic domain (Figure 3F) [27]. Interestingly, SUMO itself, the sole member of the Drosophila SUMO protein family, was identified as an interactor of the PARG disordered tail [29]. This suggests that SUMOylation may occur at both the catalytic and disordered tail regions, potentially contributing to distinct regulatory roles. Supporting this, online prediction tools identified three high-probability SUMOylation motifs in PARG: two located in the catalytic domain and one at the beginning of the disordered tail (Appendix A).

Taken together, these findings strongly suggest that PARG is regulated by SUMOylation with modifications occurring at both its catalytic and disordered tail regions. This raises the intriguing possibility that SUMOylation influences PARG stability, localization, or enzymatic activity, adding still another layer of post-translational control to PARG function.

### 2.8. The Disordered Tail Controls PARG Chromatin Binding

PARG binds to specific loci to regulate transcription [5,7], yet it lacks a DNA-binding domain, meaning it must be recruited to chromatin through protein–protein interactions. Initially, we hypothesized that PARP1 could serve as the primary recruiter of PARG to chromatin. However, our analysis revealed that PARP1 is absent from nearly 50% of PARG-bound loci, suggesting that an alternative mechanism facilitates PARG recruitment [7].

To identify additional factors involved in PARG recruitment, we again examined our mass spectrometry data, which revealed multiple chromatin-associated proteins interacting predominantly with the PARG disordered tail (Figure 3A,G). Notably, all core histones, including H2A, H2B, H3, and H4, as well as the histone variant H2Av, were among the most enriched interactors with histone H4 ranking in the 97th percentile of proteins binding to the disordered tail. Although H2A and H2B exhibited weaker, but significant, interactions with the catalytic domain, these data suggest that PARG can associate with chromatin through direct interactions with histones.

Beyond histones, we identified several chromatin remodeling and transcriptional regulatory complexes that interact specifically with the PARG disordered tail, reinforcing the idea that this domain is central to PARG recruitment. These include members of the Trithorax complex, such as Ash2, a subunit of the COMPASS complex responsible for H3K4 methylation, Mor, a member of the SWI/SNF complex, and Iswi, key member of the imitation SWI complex [30,31,32]. We also identified Mi-2, a member of the nucleosome remodeling and deacetylase (NuRD) complex, which plays a key role in chromatin accessibility and gene repression, and Caf1-55, a member of multiple complexes including Polycomb repressive complex 2 (PRC2), NuRD complex and the chromatin assembly factor 1 (CAF1) complex [33,34]. Another strong candidate for PARG recruitment is a zinc-finger protein called Protein on Ecdysone Puffs (Pep) that ranked in the 91st percentile of PARG disordered tail interactors [35].

In contrast, only two proteins that could mediate chromatin recruitment were identified as interactors of the PARG catalytic domain. PARP1 was among the strongest interactors, ranking in the 97th percentile, consistent with its known role in ADP-ribose-dependent chromatin remodeling. The second interactor was nucleophosmin (Nph), a chromatin-binding protein specifically enriched in the nucleolus, suggesting a role in PARG retention within nucleolar subdomains (Figure 3A,G) [36].

This data strongly suggests that PARG is recruited to chromatin through two distinct mechanisms. The first mechanism involves PARP1-dependent recruitment via the catalytic domain, likely through binding to pADPr chains. The second mechanism operates independently of PARP1 and is mediated by interactions between the PARG disordered tail and core histones or chromatin remodelers.

To further validate these mechanisms, we examined the subcellular localization of different PARG constructs relative to PARP1 (Figure 5). PARG-WT displayed a chromatin pattern distinct from that of PARP1, but colocalized with it at multiple loci (Figure 5A). PARG-Cat, which lacks the disordered tail, was primarily associated with the nucleolus, but remained detectable at chromatin (Figure 5B). In contrast to PARG-WT, PARG-Cat perfectly colocalizes with PARP1 (Figure 5B). PARG-D-tail, on the other hand, exhibited a chromatin-binding pattern completely exclusive of PARP1, further supporting the role of the disordered tail in PARP1-independent chromatin association (Figure 5C).

Middle panels display the corresponding plot profiles of fluorescence intensity, showing PARG constructs in green and PARP1 in red. Data are normalized from 0 to 100 to facilitate visualization of colocalization patterns. The x-axis represents distance in pixels from the starting point with each data point corresponding to the average intensity over a 15-pixel width, which matches the width of the yellow line in the top panels.

Bottom panels show dot plots illustrating the correlation between PARG and PARP1 fluorescence intensities for each construct. Each dot represents a single pixel measurement with PARG intensity on the x-axis and PARP1 intensity on the y-axis.

These data demonstrate that the PARG disordered tail is a critical determinant of chromatin targeting, enabling interactions with core histones and chromatin-remodeling complexes. In the absence of the disordered tail, PARG relies exclusively on PARP1 for chromatin binding, highlighting the functional importance of the disordered tail in maintaining proper chromatin association and activity. This dual mechanism of recruitment ensures that PARG is precisely localized to chromatin, allowing it to fine-tune transcriptional regulation in response to cellular needs.

## 3. Discussion

Many chromatin-associated proteins contain intrinsically disordered regions (IDRs), yet their functional significance remains poorly understood [1]. Despite the lack of structural resemblance in their disordered tails, we herein demonstrate that human PARG is fully functional in Drosophila, suggesting a conserved regulatory mechanism. Our data highlights that the disordered tail of PARG controls its function, with its absence rendering the protein largely cytoplasmic, non-functional, and mislocalized on chromatin. Furthermore, the disordered tail alone is sufficient to ensure correct localization, indicating that it acts as a key regulatory domain controlling nuclear-cytoplasmic trafficking and chromatin recruitment (Figure 6).

Our mass spectrometry data provide further mechanistic insights into this regulation. We found that alpha-importins, which mediate nuclear import, interact with both the catalytic and disordered tail regions, whereas the Embargoed (Emb) complex, the Drosophila ortholog of CRM1, responsible for nuclear export, interacts exclusively with the catalytic domain. This perfectly aligns with the localization patterns observed in our PARG constructs, reinforcing a model in which the disordered tail promotes nuclear retention, whereas the catalytic domain facilitates export, thus maintaining equilibrium in the system. Proteins are often exported from the nucleus as a prerequisite for degradation [21], and our data supports a link between PARG nuclear-cytoplasmic trafficking and protein stability. Specifically, we identified interactions between the PARG catalytic domain and Stub1, a cytoplasmic E3 ubiquitin ligase involved in protein degradation [25]. Additionally, we found that PARG interacts with the SCF complex, which mediates ubiquitin-dependent degradation in both the nucleus and cytoplasm [24]. This suggests that the disordered tail acts as a regulatory switch, preventing premature export and degradation, thereby ensuring proper nuclear function.

While these data establish the control of PARG nuclear–cytoplasmic trafficking by the disordered tail, our results also reveal that its function extends beyond this role. Deletion of a predicted nuclear localization signal (residues 685–710) within the disordered tail reproduces the cytoplasmic shift seen in PARG-Cat, yet, unlike PARG-Cat, this construct retains catalytic activity. Conversely, forcing the catalytic domain into the nucleus via a constitutive NLS results in aberrant subnuclear localization and loss of function. These results indicate that the disordered tail not only governs nuclear import and export but also ensures correct subnuclear targeting, which is essential for enzymatic activity. This additional role is consistent with our recent demonstration that PARG binds chromatin in two distinct patterns: at loci co-occupied with PARP1 (common loci) and at PARP1-independent loci (PARG-specific loci) [7]. Since PARG lacks a DNA-binding domain, these interactions must be mediated by other chromatin-associated factors. Our data suggest that PARP1 recruits PARG through interactions with the catalytic domain, while the disordered tail facilitates PARG-specific recruitment. Consistently, we identified direct interactions between the disordered tail and all core histones, as well as major chromatin remodeling complexes, including COMPASS, SWI/SNF, and NuRD, supporting a model in which the disordered tail directs PARG to specific genomic loci independently of PARP1.

The regulatory role of the disordered tail also appears to be integrated with post-translational modifications. We previously reported that PARG stability is regulated by phosphorylation at two sites, Ph1 and Ph2 [6]. Our mass spectrometry data now reveals that the kinases responsible for phosphorylating these sites are distinct in that GSK3/Shaggy (Sgg) interacts preferentially with the catalytic domain where Ph1 is located, whereas CKIIalpha is enriched on the disordered tail where Ph2 resides. Since both kinases are active in nuclear and cytoplasmic compartments, the precise spatial regulation of PARG phosphorylation remains an open question.

Post-translational modifications may provide an additional layer of regulation. Our data reveals a strong interaction between PARG and SUMOylation machinery, particularly with Nup358, a nuclear pore-associated E3 SUMO ligase known to SUMOylate proteins entering the nucleus [28]. Additionally, SUMO itself interacts specifically with the disordered tail, and online prediction tools identified SUMOylation sites in both the catalytic and disordered tail regions. SUMOylation could serve two possible roles. It may either stabilize PARG by preventing nuclear exports or compete with ubiquitination to protect PARG from degradation [37,38].

Collectively, these results establish the disordered tail region as a central hub for PARG regulation, integrating nuclear-cytoplasmic trafficking, stability, post-translational modifications, and chromatin recruitment. Therefore, this study provides a comprehensive model that explains the dynamic regulation of PARG within its intrinsically disordered regions, revealing new mechanisms that could have broader implications for chromatin-associated IDR-containing proteins.

## 4. Materials and Methods

Drosophila Strains and Genetics. All flies were maintained at 20 °C unless otherwise specified. The *parg^27.1^* null allele was described in [39] and the GawB(69B)-Gal4 driver in [40]. The following genotypes were used:
PARG^WT^: *parg^27.1^*/*parg^27.1^*; P{w^+^, UASt::PARG-WT-EGFP}, P{w^+^, UASt::PARP1-DsRed}, P{w^+^, *Gawb(69b)*-Gal4}/*TM2*.Human PARG (hPARG): *parg^27.1^*/*parg^27.1^*; P{w^+^, UASt::hPARG-EGFP}, P{w^+^, UASt::PARP1-DsRed}, P{w^+^, *Gawb(69b)*-Gal4}/*TM2*.PARG^cat^: *parg^27.1^*/*parg^27.1^*; P{w^+^, UASt::PARG-Cat-EGFP}, P{w^+^, UASt::PARP1-DsRed}, P{w^+^, *Gawb(69b)*-Gal4}/*TM2*.PARG^Cter^: *parg^27.1^*/*FM7i-GFP*; P{w^+^, UASt::PARG-D-tail-EGFP}, P{w^+^, UASt::PARP1-DsRed}, P{w^+^, *Gawb(69b)*3Gal4}/*Tm2*.

Because PARG-D-tail does not rescue the developmental arrest caused by the *parg^27.1^* mutation, this stock is maintained in a heterozygous state. Homozygous *parg^27.1^* mutant third instar larvae expressing PARG-D-tail were identified by the absence of the FM7i-GFP balancer marker.

Generation of PARG Constructs. All PARG constructs were cloned into the pUASt-EGFP vector for expression in vivo.

PARG-WT was amplified by PCR from genomic DNA using the following primers:
Forward: AAAGCGGCCGCATGCAAGAATTCAGGTCACACTTGReverse: AAAGGTACCGTCAACAATATCGTCCTTTTCGTC

This amplicon includes the full-length PARG coding sequence along with its native 5’ UTR and intronic regions.

Human PARG (hPARG) was subcloned from the hPARG-EGFP-N3 vector described in [41] into the pUASt-EGFP backbone.

PARG-Cat was amplified by PCR from genomic DNA using the following primers:Forward: AAGGTACCATGCAAGAATTCAGGTCACACTTGReverse: TTGGATCCGGCGGATGCTCCCTC

This amplicon encodes the N-terminal catalytic domain of PARG (residues 1–606) and includes the 5′ UTR and intronic regions.

The PARG-D-tail was amplified by PCR from genomic DNA using the following primers:Forward: AAGAATTCATGGAAGCTGGAAGCTCTAGAGReverse: AAGGTACCGTCAACAATATCGTCCTTTTCGTC

This fragment encodes the disordered tail of PARG (residues 607–768) and includes an in-frame ATG at the 5′ end to initiate translation.

PARG-Δ685–710, lacking residues 685–710 (predicted to form a nuclear localization signal, NLS [6]), was generated by amplifying PARG using reverse primers that skip the targeted region, followed by cloning into the pUASt-EGFP vector.Forward primer: AAAGCGGCCGCATGCAAGAATTCAGGTCACACTTGReverse primer: AAAGGTACCGTCAACAATATCGTCCTTTTCGTCCTTGTCGGTCACATCCTTTTCATAATGG

PARG-Cat-cNLS was generated by amplifying the PARG-Cat fragment (residues 1–606) from the corresponding plasmid using:Forward primer: AAGCGGCCGCATGCAAGAATTCAGGTCACACTTGReverse primer: TTGGTACCGGCGGATGCTCCCTCTC

This fragment was cloned into the pUASt-EGFP-cNLS vector containing a constitutive nuclear localization signal (cNLS). The cNLS sequence used (RDPKKKRKVDPKKKRKVDPKKKRKV) is sufficient to drive strong nuclear localization, as previously demonstrated [11].

Prediction of Intrinsically Disordered Region (IDRs). IDR predictions for human and Drosophila PARG were performed using the MobiDB online platform [16]. MobiDB integrates outputs from multiple predictive models; for this analysis, we relied on IDR annotations from both MobiDB-lite and AlphaFold-based predictions to assess the disordered regions across each protein.

Protein Structure Prediction and Structural Alignment. As only the catalytic domains of human and Drosophila PARG have been resolved by crystallography, full-length molecular structure predictions were generated using the Chai-1 deep learning-based prediction tool, v0.5.2 (chaidiscovery.com, accessed on 10 January 2025). Predicted models with the highest confidence scores were selected and structurally aligned using PyMOL v3.1.0. The structural alignment and corresponding rotation movie were also generated with PyMOL v3.1.0.

Adult Lifespan Measurement. To compare the adult lifespan of flies expressing PARG-WT, hPARG, or PARG-Cat, 20 virgin females and 20 virgin males were placed in vials containing standard cornmeal-molasses-agar medium supplemented with dry yeast. Flies were maintained at 25 °C and transferred to fresh food every two days. Dead flies were counted daily at fixed times. Each condition was tested in duplicate.

Developmental Timing Measurement. For each condition, 10 virgin females and 10 virgin males were placed in fresh food vials and allowed to mate and lay eggs for 72 h. Adults were then removed, and the vials were monitored daily at fixed times. The number of pupae was recorded each day to assess developmental progression. The experiment was performed in triplicate.

Western blotting. The following reagents and antibodies were used for immunoblotting: anti-poly(ADP-ribose) (pADPr) detection reagent (1:6000; MABE1031, Sigma, Ronkonkoma, NY, USA), anti-GFP antibody (mouse monoclonal, 1:4000; JL-8, Takara, San Jose, CA, USA, BD #632380), and anti-Histone H3 antibody (rabbit monoclonal, 1:4000; Abcam, Burlingame, CA, USA, ab176842). Secondary antibodies included goat anti-mouse (1:5000; Invitrogen, Waltham, MA, USA, G-21040) and goat anti-rabbit (1:5000; Revvity, Waltham, MA, USA, NEF812001EA).

Signal detection was performed using the Pierce™ ECL Western blotting Substrate (Thermo Scientific, Waltham, MA, USA, 32106), according to the manufacturer’s instructions.

Nuclear/Cytoplasm Fractionation. 10 wandering third instar larvae expressing PARG-Cat were collected and briefly washed in PBS. Larvae were homogenized on ice in 200 µL of nuclear extraction buffer (5 mM MgCl_2_, 10 mM HEPES, 0.5% Triton X-100, 320 mM sucrose) supplemented with 1X protease inhibitor cocktail (Pierce, Thermo Scientific, Waltham, MA, USA). The homogenate was transferred into a cut 0.5 mL Eppendorf tube lined with Calbiochem filter paper and placed inside a 1.5 mL Eppendorf tube. Samples were centrifuged at 2000× *g* for 5 min at 4 °C.

The flowthrough was collected, gently vortexed, and incubated on ice for 10 min. It was then centrifuged again at 2000× *g* for 5 min at 4 °C. The resulting supernatant, representing the cytoplasmic fraction, was mixed 1:1 with Laemmli buffer and boiled at 99 °C for 3 min. The pellet, containing the nuclear fraction, was washed twice with 500 µL of nuclear washing buffer (5 mM MgCl_2_, 10 mM HEPES, 320 mM sucrose) for 5 min each and centrifuged at 2000× *g* for 5 min at 4 °C. The final pellet was resuspended in 40 µL Laemmli buffer and 40 µL PBS, then incubated at 99 °C for 3 min.

Sample Preparation for Imaging. Wandering third instar larvae were dissected in Grace’s medium on a concave slide. Salivary glands, brains, and wing or eye imaginal disks were collected and fixed for 30 min at room temperature on a rotating wheel in fixation buffer (2% paraformaldehyde, 0.1% Triton X-100 in PBS). Samples were washed twice for 5 min in PBST (0.1% Triton X-100 in PBS), then incubated with RNase A (0.5 µg/µL; Qiagen, Germantown, MD, USA, #19101) for 20 min at room temperature and washed once in PBS. DNA was stained with TOTO-3 (1:4000; Invitrogen, Waltham, MA, USA, T3604) for 20 min in PBST at room temperature and washed once more in PBST for 5 min. Samples were then mounted in Vectashield mounting medium (Vector Laboratories, Burlingame, CA, USA).

For embryo preparation, adult flies were placed in a cage overnight on a fruit agar plate (25% apple juice, 2.5% sucrose, 2.5% agar, 0.03% Tegosept) topped with fresh yeast paste. Embryos were collected using a mesh, rinsed with water, and dechorionated for 2 min in 10% bleach, followed by vigorous washing in water. Fixation was carried out for 20 min in a solution containing 50% heptane, 0.1% Triton X-100, 1X PBS, and 1.85% paraformaldehyde. After fixation, the lower aqueous phase was removed, and an equal volume of methanol was added. Samples were shaken vigorously for 30 s to rupture the vitelline membrane. Embryos at the interface were collected, washed twice in methanol and once in PBST, and mounted in Vectashield.

Immunoprecipitation. Immunoprecipitation was performed using four different Drosophila lines: PARG-Wt, PARG-Cat, PARG-D-tail and *y^1^*, *w^1118^* flies (*yw*), the latter serving as a negative control that does not express GFP. All experiments were carried out in duplicate. For each condition, approximately 300 females and 300 males were placed overnight in 1 L egg-laying cages sealed with juice agar plates (25% apple juice, 2.5% sucrose, 2.5% agar, 0.03% Tegosept) and topped with fresh yeast paste. The following day, 150 mg of embryos were collected per cage and homogenized on ice using a pre-chilled homogenizer in 1 mL of default lysis buffer (DLB: 50 mM Tris-HCl pH 7.5, 5% glycerol, 0.2% IGEPAL, 1.5 mM MgCl_2_, 125 mM NaCl, 125 mM NaF, 1 mM Na_3_VO_4_, 1 mM DTT) supplemented with cOmplete Protease Inhibitor Cocktail (Roche, Basel, Switzerland, 11697498001). Lysates were incubated on ice for 20 min and centrifuged at 16,000× *g* for 20 min at 4 °C. The supernatant was transferred to a fresh tube and centrifuged again under the same conditions before being flash-frozen in liquid nitrogen and stored at −80 °C.

Upon thawing on ice, samples were again centrifuged at 16,000× *g* for 20 min at 4 °C. To reduce non-specific binding, lysates were pre-cleared by incubating with 100 µL of 50% slurry Protein A agarose beads (Pierce, 20421) for 1 h at 4 °C, followed by centrifugation at 500× *g* for 1 min at 4 °C. The supernatant was filtered through a 0.45 µm syringe filter into a pre-chilled tube. Filtered lysates were incubated with 40 µL of 50% slurry GFP-Trap agarose beads (ChromoTek, Rosemont, IL, USA, gta-20) for 2 h at 4 °C with gentle rotation. Beads were pelleted at 500× *g* for 1 min at 4 °C, and washed four times with 1 mL of DLB for 5 min each at 4 °C, followed by centrifugation at 500× *g*. Bound proteins were eluted by resuspending the beads in 40 µL of 2× Laemmli buffer supplemented with 2.5 µL of 1 M DTT and 10 µL of 10% SDS, followed by incubation at 99 °C for 5 min. Protein recovery was confirmed by silver staining, and the presence of PARG-GFP bait was verified by Western blot using an anti-GFP antibody (mouse monoclonal, 1:4000; JL-8, Takara, San Jose, CA, USA, BD #632380).

Mass Spectrometry Analysis. Samples were separated on a 4–12% gradient Bis-Tris acrylamide gel and run approximately 1 cm into the resolving gel to concentrate the proteins. Gels were stained with Coomassie Brilliant Blue, and each lane was excised in its entirety. Protein identification was carried out by the Taplin Mass Spectrometry Facility using liquid chromatography coupled to tandem mass spectrometry (LC-MS/MS).

To identify specific PARG interactors, spectral count data were analyzed using the SAINT (Significance Analysis of INTeractome) algorithm v2.5.0, comparing experimental samples to the *yw* negative control [42]. Proteins were considered high-confidence interactors if they had a SAINT score greater than 0.5 and a false discovery rate (FDR) below 0.15. For each high-confidence interactor, a log2 fold change (log2FC) was calculated based on spectral counts from PARG-Cat and PARG-D-tail samples. A negative log2FC indicates preferential interaction with the catalytic domain, whereas a positive value indicates preferential interaction with the C-terminal domain.

Colocalization Analysis. Colocalization between PARG-GFP and PARP1-DsRed was analyzed using Fiji (ImageJ, v1.54). For each sample, fluorescence intensity was measured point-by-point along a defined line spanning a chromosomal region, and the average pixel intensity for both channels was recorded at each position. Data are presented as fluorescence intensity histograms and as scatterplots comparing the signal from PARG-GFP and PARP1-DsRed. The degree of colocalization was quantified by calculating the Pearson correlation coefficient from the scatterplot data.

## Figures and Tables

**Figure 1 ijms-26-08166-f001:**
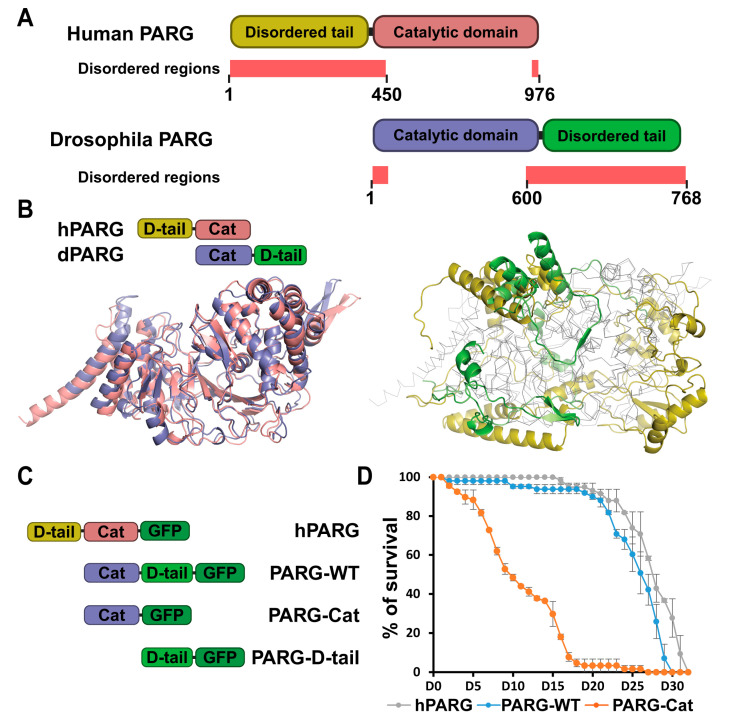
Disordered tails are structurally different between human and drosophila PARG. (**A**) Schematic representation of human and Drosophila PARG proteins. Human PARG (hPARG) contains a unique 360-amino acid N-terminal disordered tail that is absent in Drosophila. Conversely, Drosophila PARG (dPARG) features a distinct 161-amino acid C-terminal disordered tail that is not present in hPARG. Disordered region predicted by MobiDB and AlphaFold are highlighted in light red (See Section 4). The disordered region spans residues 1–370 for human PARG and residues 600–768 for drosophila PARG (**B**) The left panel displays the structural alignment of the catalytic domains of hPARG (pink) and dPARG (blue), highlighting their strong conservation. The right panel displays the structural alignment of the non-catalytic domains: the N-terminal disordered tail (D-tail) of hPARG (yellow) and the C-terminal disordered tail of dPARG (green). The catalytic domains are shown in gray outlines to emphasize structural divergence between the terminal regions. (**C**) Schematic representations of GFP-tagged constructs of hPARG, PARG-WT, PARG-Cat, or PARG-D-tail. hPARG and PARG-WT represent the full-length Human and Drosophila PARG, respectively. PARG-Cat includes only the catalytic domain (residues 1–562), excluding the C-terminal region, while PARG-D-tail includes only the C-terminal disordered tail (residues 563–723), excluding the catalytic domain. (**D**) Lifespan curves of adult flies expressing hPARG (gray), PARG-WT (blue), or PARG-Cat (orange) in a *parg* mutant background. The y-axis represents the percentage of surviving flies with Day 0 corresponding to adult emergence. Data represents the average of triplicate experiments, and error bars indicate the standard error of the mean (SEM).

**Figure 2 ijms-26-08166-f002:**
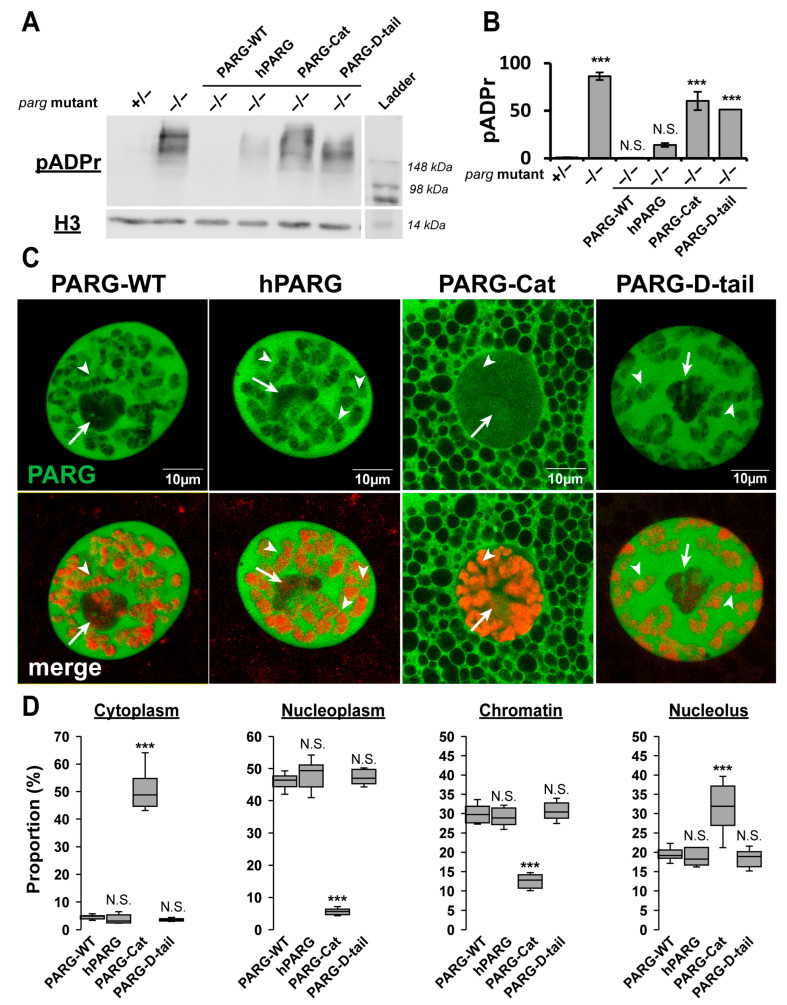
The Disordered Tail Controls PARG Activity and Subcellular Localization. (**A**) Western blot analysis of pADPr levels (top panel), and Histone H3 as a loading control (bottom panel) in transgenic flies expressing PARG-WT, hPARG, PARG-Cat, or PARG-D-tail in a *parg* null mutant background (−/−). Heterozygous *parg* mutant (+/−) and homozygous *parg* mutant (−/−) serve as controls for low and high pADPr levels, respectively. The last lane contains the molecular weight ladder. (**B**) Quantification of pADPr levels, averaged from three independent blots. Statistical analysis was performed using a two-tailed *t*-test with heterozygous *parg* mutant as the reference. *** indicates *p* < 0.01, and N.S. denotes non-significant results. (**C**) Representative images of a single nucleus from late third instar larval salivary gland cells, a polytenized tissue. The upper panel shows GFP-tagged PARG signals in transgenic lines expressing PARG-WT, hPARG, PARG-Cat, or PARG-D-tail (left to right). The lower panel displays merged signals with DNA stained in red. White arrows mark the nucleolus, and arrowheads highlight representative examples of PARG association with chromatin. (**D**) Quantification of PARG protein distribution across cellular compartments: cytoplasm, nucleoplasm, chromatin, and nucleolus. Data represents the average signal intensity measured across ten independent images per transgenic line. Statistical analysis was performed using a two-tailed t-test with PARG-WT as the reference. *** indicates *p* < 0.01, and N.S. denotes non-significant results.

**Figure 3 ijms-26-08166-f003:**
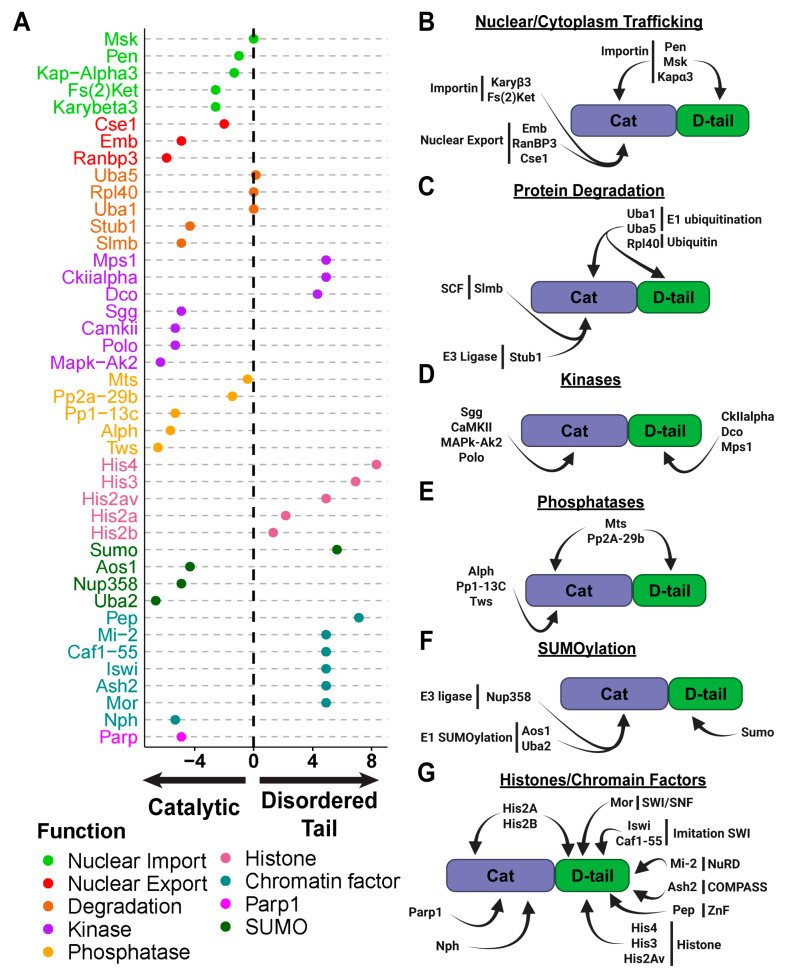
The Disordered Tail Mediates PARG Protein–Protein Interactions Across Diverse Functional Pathways. (**A**) Dot chart summarizing proteins identified as interactors of PARG based on mass spectrometry data. The y-axis lists the identified proteins, while the color code indicates their primary functional category. The x-axis represents the log2 fold change in interaction preference between PARG-Cat (catalytic domain only) and PARG-D-tail (disordered tail only) (see Section 4). Positive values indicate preferential binding to the disordered tail, while negative values indicate preferential binding to the catalytic domain. Values near zero suggest no significant bias toward either domain. (**B**–**G**) Categorized breakdown of proteins interacting with specific domains of PARG. For each functional category, proteins are grouped based on their interaction specifically with the catalytic domain, the disordered tail, or both, highlighting the distinct roles of each domain in protein–protein interactions.

**Figure 4 ijms-26-08166-f004:**
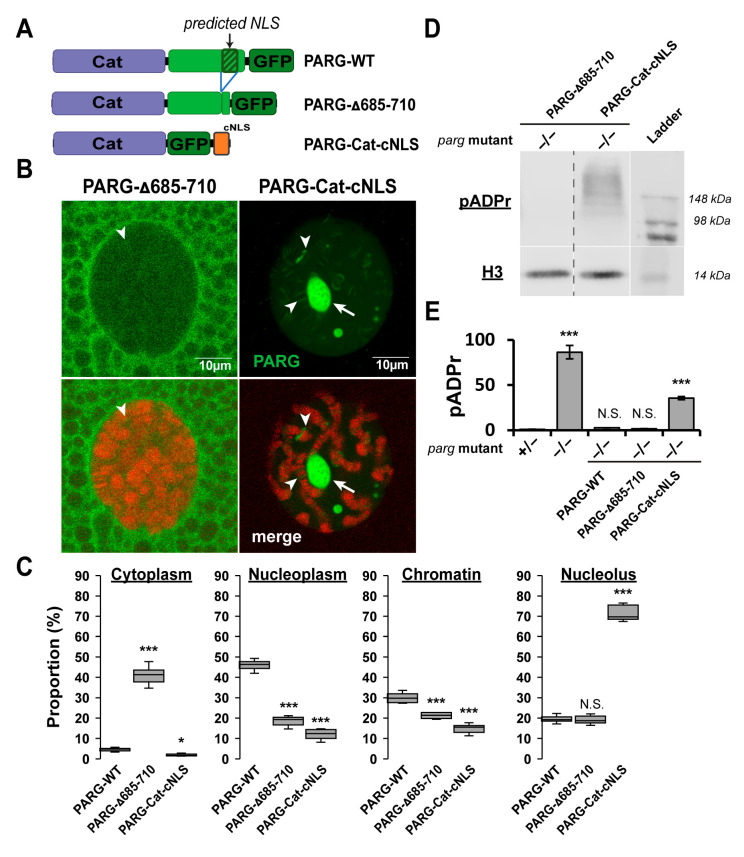
The Disordered Tail Controls PARG Nuclear–Cytoplasmic Trafficking and Functional Nuclear Localization. (**A**) Schematic representation of GFP-tagged PARG constructs: PARG-WT, PARG-Δ685–710, and PARG-Cat-cNLS. PARG-Δ685–710 is full-length PARG lacking residues 685–710, a region predicted to contain a nuclear localization signal (NLS) [6]. PARG-Cat-cNLS lacks the C-terminal disordered tail (residues 563–723) but contains a constitutive NLS (see Section 4). (**B**) Representative confocal images of single nuclei from late third instar larval salivary gland cells (polytenized tissue). Upper panels show PARG-GFP signals in the indicated transgenic lines; lower panels show merged GFP (green) and DNA (red) signals. White arrows mark the nucleolus, and arrowheads highlight representative examples of PARG association with chromatin. (**C**) Quantification of PARG signal distribution across cytoplasm, nucleoplasm, chromatin, and nucleolus. Data represents mean signal intensity from ten independent images per genotype. Statistical analysis was performed using a two-tailed *t*-test with PARG-WT as the reference. * *p* < 0.05, *** *p* < 0.01; N.S., not significant. (**D**) Western blot analysis of pADPr levels (top) and Histone H3 (bottom; loading control) in transgenic flies expressing PARG-Δ685-710 or PARG-Cat-cNLS in a *parg* null mutant background (−/−). The last lane contains the molecular weight ladder. (**E**) Quantification of pADPr levels from three independent blots. Statistical analysis was performed using a two-tailed *t*-test with *parg* heterozygous mutants as the reference (see Figure 2). *** *p* < 0.01; N.S., not significant.

**Figure 5 ijms-26-08166-f005:**
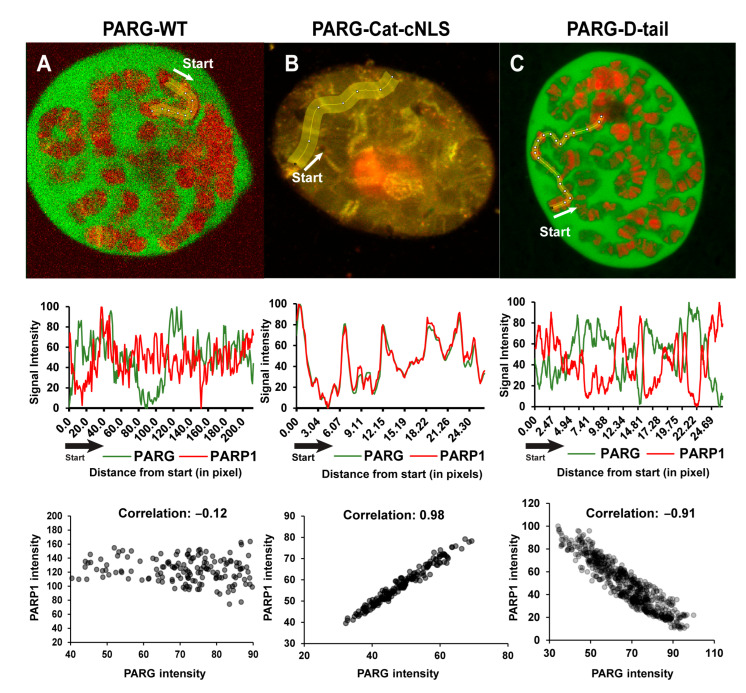
The Disordered Tail Controls PARG Chromatin Binding. (**A**–**C**) Colocalization analysis between PARP1 and different PARG constructs: PARG-WT (**A**), PARG-Cat-cNLS (**B**), and PARG-D-tail (**C**). Top panels show representative images of PARG constructs (green) and PARP1 (red). A yellow line marks the region used for fluorescence intensity measurements with the start point indicated and the arrow showing the direction of the measurement. Since the nuclear fraction of PARG-Cat is too low for reliable quantification, colocalization was measured using PARG-Cat-cNLS, a nuclear-localized version of PARG-Cat (see Section 4).

**Figure 6 ijms-26-08166-f006:**
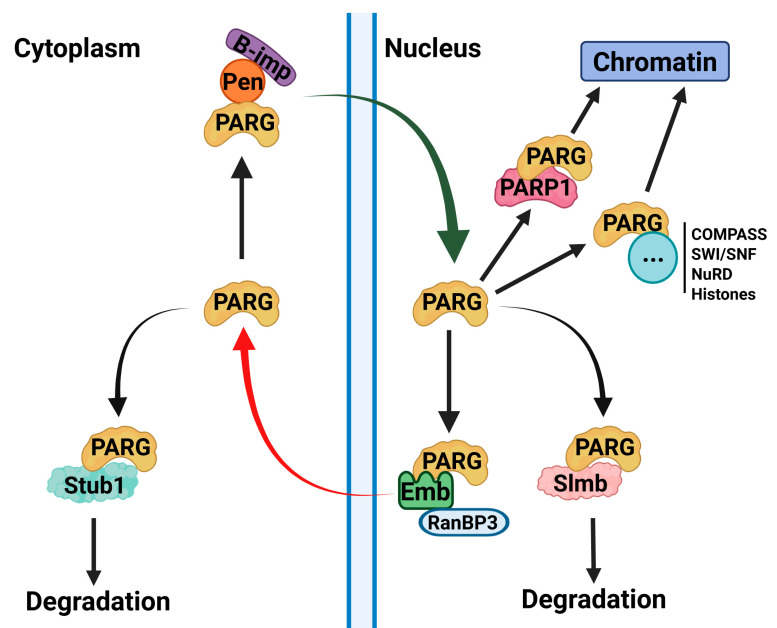
Mechanisms of PARG Regulation. This model summarizes the regulatory mechanisms controlling PARG localization, stability, and chromatin recruitment suggested by the data presented in this study. Our findings suggest that PARG nuclear import is mediated by its interaction with Pendulin, whereas its nuclear export is facilitated by the Emb complex. Upon export, PARG may be targeted for degradation through its interaction with the E3 ubiquitin ligase Stub1. However, SUMOylation or phosphorylation of PARG may inhibit this export, thereby stabilizing the protein. Alternatively, SUMOylation or phosphorylation could act within the nucleus to prevent Slmb-dependent degradation, providing an additional layer of regulation. In terms of chromatin recruitment, we identified two distinct mechanisms: (1) PARP1-dependent recruitment whereby PARG interacts with PARP1 through its catalytic domain and co-occupies shared loci and (2) PARP1-independent recruitment which may occur through interactions with chromatin remodeling complexes, such as COMPASS, SWI/SNF, and NuRD, or through direct nucleosome binding, all of which are mediated by the PARG disordered tail.

## Data Availability

All transgenic lines and reagents generated in this study are available upon request.

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
