# Peer review of "Disordered Protein Tail Is Wagging Poly(ADP-ribosyl)ation"

_ijms, 2025, doi:10.3390/ijms26178166_

Round 1
Reviewer 1 Report
Comments and Suggestions for Authors
Intrinsically disordered regions are crucial in cell signaling and transcription because of their flexible binding and interactions. The authors in this study explored the role of the disordered tail region in Poly(ADP-ribose) glycohydrolase (PARG) regulation in Drosophila. They found that removing the tail region of PARG caused its mislocalization in the cytoplasm, disrupted nuclear import and export, and impaired PARG's association with chromatin. They showed that the removal of the disordered C-terminal tail in PARG caused developmental delay and reduced lifespan in Drosophila. So, it is crucial to understand the PARG regulation, as it’s involved with a variety of regulatory activities, including nuclear-cytoplasmic trafficking, chromatin recruitment, and DNA repair. IP-MS based studies identified a total of 387 proteins that significantly interact with PARG. The authors identified the specific domains in PARG responsible for interactions with nuclear transport factors, enzymes, and chromatin complexes. For example, the authors claim that alpha and beta-importins interact with both the catalytic and disordered tail regions of PARG, while Emb and RanBP3 interact with only the catalytic domain, which helps maintain the equilibrium between nuclear import and export of PARG. However, these interactions should be validated to strengthen these claims. For example, the authors could employ a pull-down assay to verify these interactions. Similarly, in Figure S4, the two distinct bands were observed for PARG-Cat, and the authors attribute the 2nd band to the Sumoylation of PARG-Cat. These findings could be validated by mutating the lysines in the SUMOylation regions and performing a Western blot.
Some minor concerns:
- Figure 1 was mislabeled; only Figures A-D exist, but E was mentioned in the caption
- Line 144: Should be Fig 1C
- Include the molecular weight ladder in the figures.
- Lines 313-316: Nucleophosmin (Nph) was not shown in Fig. 3A.
Overall, the manuscript was well written. I suggest that the authors address the above concerns before publishing the paper.
Author Response
R1: Intrinsically disordered regions are crucial in cell signaling and transcription because of their flexible binding and interactions. The authors in this study explored the role of the disordered tail region in Poly(ADP-ribose) glycohydrolase (PARG) regulation in Drosophila. They found that removing the tail region of PARG caused its mislocalization in the cytoplasm, disrupted nuclear import and export, and impaired PARG's association with chromatin. They showed that the removal of the disordered C-terminal tail in PARG caused developmental delay and reduced lifespan in Drosophila. So, it is crucial to understand the PARG regulation, as it’s involved with a variety of regulatory activities, including nuclear-cytoplasmic trafficking, chromatin recruitment, and DNA repair. IP-MS based studies identified a total of 387 proteins that significantly interact with PARG. The authors identified the specific domains in PARG responsible for interactions with nuclear transport factors, enzymes, and chromatin complexes. For example, the authors claim that alpha and beta-importins interact with both the catalytic and disordered tail regions of PARG, while Emb and RanBP3 interact with only the catalytic domain, which helps maintain the equilibrium between nuclear import and export of PARG. However, these interactions should be validated to strengthen these claims. For example, the authors could employ a pull-down assay to verify these interactions.
Response: We thank Reviewer 1 for their careful reading of our work. In the revised version of our manuscript, we investigated further the role of the disordered tail in nuclear–cytoplasmic trafficking, we assessed the functional importance of the predicted nuclear localization signal (residues 685–710). Specifically, we examined the effects of NLS deletion (PARG-Δ685–710) and of forcing nuclear import of the catalytic domain via a constitutive NLS (PARG-Cat-cNLS). These experiments demonstrate that the disordered tail controls not only PARG nuclear–cytoplasmic trafficking but also its subnuclear localization to ensure the correct enzymatic activity. The Results and Discussion sections have been updated accordingly to incorporate and clarify these statements.
R1: Similarly, in Figure S4, the two distinct bands were observed for PARG-Cat, and the authors attribute the 2nd band to the Sumoylation of PARG-Cat. These findings could be validated by mutating the lysines in the SUMOylation regions and performing a Western blot.
Response: We appreciate this important comment. While direct biochemical confirmation of the interaction between the disordered tail and the SUMOylation machinery would be valuable, it is beyond the scope of this study. Our work focuses on uncovering a novel regulatory mechanism of PARG mediated by its intrinsically disordered tail. As shown in Figures 1–5, deletion of this region disrupts localization, impairs chromatin association, reduces catalytic activity, and decreases organismal viability. To further characterize its regulatory network, we performed mass spectrometry, identifying 289 proteins specifically associated with the catalytic domain and 191 with the disordered tail. Gene ontology analyses of these interactors align perfectly with the phenotypes observed upon deletion of either domain; notably, both domains are essential for precise chromatin binding (Fig. 5). The SUMOylation machinery constitutes only one subset within the broader network of proteins interacting with the disordered tail.
R1: Figure 1 was mislabeled; only Figures A-D exist, but E was mentioned in the caption
Line 144: Should be Fig 1C.
Response: We thank the reviewer for pointing this out. We have corrected the figure legend and the text in line 144 to accurately refer to Figure 1C.
R1: Include the molecular weight ladder in the figures.
Response: The molecular weight ladder has now been included in Figure 2A.
R1: Lines 313-316: Nucleophosmin (Nph) was not shown in Fig. 3A.
Overall, the manuscript was well written. I suggest that the authors address the above concerns before publishing the paper.
Response: We have revised Figure 3A to now include Nucleophosmin (Nph) as referenced in the text. We are grateful for the reviewer’s careful reading and constructive feedback.
Reviewer 2 Report
Comments and Suggestions for Authors
The manuscript titled "Disordered Protein Tail is Wagging Poly(ADP-ribosyl)ation" presents a systematic analysis of the IDR in the PARG protein and give the verification of several post-translational modifications. This study is based on a reasonably analysis around PARG. However, the novelty of this paper is limited and the previous studies from the same group have already suggested roles for PARG in gene regulation and localization. Therefore, I recommend a major revision before the manuscript can be considered for publication.
Major Concerns:
1. The section 2.5 and 2.6 proposed the effects under SUMOlyation, phosphorylation, and ubiquitination. While the mass spectrometry data and motif predictions suggest post-translational modifications in PARG at both the catalytic and disordered regions, there is no direct experimental validation of these modifications. The mutagenesis at the identified SUMOylation or phosphorylation sites (e.g., K→R or S→A substitutions) is needed to confirm the regulatory roles.
2. The section 2.2 and 2.3 presented the subcellular localization is mainly controle by the C-terminal trail region. The PARG-D-tail does not rescue the phenotype, which is expected due to loss of catalytic function, but this raises the question: Is localization alone sufficient for any aspect of PARG’s function (e.g., chromatin binding)? I would suggest to the authors to test whether the disordered tail localizes to chromatin in the absence of the catalytic domain.
3. The disordered tail is an important part to show the regulation. However, there is no investigation into whether the tail contains functional motifs (e.g., NLS, NES, SUMO sites) or interacts directly with nuclear transport components. I would suggest to first report the full sequence and the potential motifs as a separate figure, then conduct motif deletion or truncation experiments and test whether nuclear import/export is perturbed (e.g., using importin/exportin inhibitors or NLS prediction/mutation).
Author Response
R2: The manuscript titled "Disordered Protein Tail is Wagging Poly(ADP-ribosyl)ation" presents a systematic analysis of the IDR in the PARG protein and give the verification of several post-translational modifications. This study is based on a reasonably analysis around PARG. However, the novelty of this paper is limited and the previous studies from the same group have already suggested roles for PARG in gene regulation and localization. Therefore, I recommend a major revision before the manuscript can be considered for publication.
Response: We thank Reviewer 2 for their thoughtful evaluation of our manuscript. This study presents the first identification of a regulatory mechanism acting on PARG through its intrinsically disordered tail, an entirely novel finding that has not been previously reported. Given our recent discovery that PARG controls transcriptional regulation independently of PARP1 (PMID: 37985852, or reference 7 in the manuscript), uncovering how regulating PARG itself is important. This is the first study that defines a mechanism that regulates PARG activity and localization independently of PARP1.
R2: The section 2.5 and 2.6 proposed the effects under SUMOlyation, phosphorylation, and ubiquitination. While the mass spectrometry data and motif predictions suggest post-translational modifications in PARG at both the catalytic and disordered regions, there is no direct experimental validation of these modifications. The mutagenesis at the identified SUMOylation or phosphorylation sites (e.g., K→R or S→A substitutions) is needed to confirm the regulatory roles.
Response: We thank Reviewer 2 for this suggestion. We previously identified and studied PARG phosphorylation sites, including mutagenizing the phosphorylated serine to alanine (phospho-mutant) or glutamic acid (phospho-mimicking) (see PMID: 34949666 or reference 6 in the manuscript).
Regarding SUMOylation, our work focuses on uncovering a novel regulatory mechanism of PARG mediated by its intrinsically disordered tail. As shown in Figures 1–5, deletion of this region disrupts localization, impairs chromatin association, reduces catalytic activity, and decreases organismal viability. To further characterize its regulatory network, we performed mass spectrometry, identifying 289 proteins specifically associated with the catalytic domain and 191 with the disordered tail. Gene ontology analyses of these interactors align perfectly with the phenotypes observed upon deletion of either domain; notably, both domains are essential for precise chromatin binding (Fig. 5). The SUMOylation machinery constitutes only one subset within the broader network of proteins interacting with the disordered tail, and while preliminary evidence suggests its involvement, direct biochemical confirmation of SUMOylation will be the focus of future studies.
R2: The section 2.2 and 2.3 presented the subcellular localization is mainly controle by the C-terminal trail region. The PARG-D-tail does not rescue the phenotype, which is expected due to loss of catalytic function, but this raises the question: Is localization alone sufficient for any aspect of PARG’s function (e.g., chromatin binding)? I would suggest to the authors to test whether the disordered tail localizes to chromatin in the absence of the catalytic domain.
Response: We thank Reviewer 2 for this insightful comment. As shown in Figure 2, the absence of the disordered tail impairs both the localization and catalytic activity of PARG. This is further supported by the drastic reduction in lifespan observed in flies expressing the tail-deleted variant (Figure 1D). Importantly, loss of the disordered tail abolishes the characteristic chromatin-binding pattern of PARG, which becomes indistinguishable from PARP1 localization (Figure 5), despite PARG normally binding to a subset of loci independently of PARP1. Altogether, these results collectively demonstrate that the disordered tail is essential not only for localization but also for chromatin association and catalytic function.
R2: The disordered tail is an important part to show the regulation. However, there is no investigation into whether the tail contains functional motifs (e.g., NLS, NES, SUMO sites) or interacts directly with nuclear transport components. I would suggest to first report the full sequence and the potential motifs as a separate figure, then conduct motif deletion or truncation experiments and test whether nuclear import/export is perturbed (e.g., using importin/exportin inhibitors or NLS prediction/mutation).
Response: We thank Reviewer for this valuable suggestion. We previously published a full sequence alignment between human and Drosophila PARG, including all predicted domains (PMID: 34949666; reference 6 in the manuscript, Supplemental Fig. S1). In the revised manuscript, we have added a new Supplemental Figure S2 showing sequence alignments of the PARG disordered tail from human, Drosophila, C. elegans, and D. rerio.
We have also incorporated new experimental data in Figure 4 assessing the functional importance of the predicted nuclear localization signal (residues 685–710) within the disordered tail. Specifically, we examined the effects of NLS deletion (PARG-Δ685–710) and of forcing nuclear import of the catalytic domain via a constitutive NLS (PARG-Cat-cNLS). These experiments demonstrate that the disordered tail controls not only PARG nuclear–cytoplasmic trafficking but also for controls the correct subnuclear localization of PARG, ensuring full enzymatic activity.
The Results and Discussion sections have been updated accordingly to incorporate and clarify these findings. We again thank Reviewer 2 for the constructive feedback, which has significantly strengthened our manuscript.
Reviewer 3 Report
Comments and Suggestions for Authors
This manuscript investigates the regulatory role of the IDR tail of dPARG and presents evidence that this region is critical for nuclear localization, chromatin association, and enzymatic activity. The authors combine transgenic models with MS-based interactomics to identify interaction partners and propose a model in which the tail acts as a hub for nuclear import, subnuclear localization, and protein turnover. The study is clearly written and presents compelling data.
Comments:
- The comparison between Drosophila and mammalian PARG domains provides some insight, but it may not be sufficient to draw solid conclusions about domain conservation. It would strengthen the evolutionary perspective to include additional species in the domain alignment to determine whether a disordered tail is broadly conserved across metazoans.
- While the authors emphasize the divergence between human and Drosophila PARG tail sequences, a direct sequence comparison is not shown. It would be useful to identify whether short functional motifs (e.g., nuclear localization signals, SUMOylation sites) are present in both species despite low overall similarity. A functional domain swapping experiment,replacing the Drosophila tail with the human counterpart and testing for rescue of nuclear localization or chromatin association, could provide stronger evidence for conserved functionality.
- The proposed role of the tail in regulating nuclear import and export is plausible based on the MS data and subcellular localization patterns. Mutagenesis of predicted importin-binding motifs (e.g., PY-NLS-like elements) and testing their effect on nuclear localization would help determine whether importin recognition is essential.
- The identification of SUMO pathway components in the interactome is intriguing but not fully explored. Mutation of candidate SUMOylation sites (e.g., K-to-R substitutions) and subsequent analysis of PARG localization or stability would help clarify.
- The proposed model in which cytoplasmic Stub1 mediates degradation of PARG is only based on MS data. Including cycloheximide chase assays comparing the degradation kinetics of PARG-WT and PARG-Cat would provide more direct support for this mechanism.
Author Response
R3: This manuscript investigates the regulatory role of the IDR tail of dPARG and presents evidence that this region is critical for nuclear localization, chromatin association, and enzymatic activity. The authors combine transgenic models with MS-based interactomics to identify interaction partners and propose a model in which the tail acts as a hub for nuclear import, subnuclear localization, and protein turnover. The study is clearly written and presents compelling data.
The comparison between Drosophila and mammalian PARG domains provides some insight, but it may not be sufficient to draw solid conclusions about domain conservation. It would strengthen the evolutionary perspective to include additional species in the domain alignment to determine whether a disordered tail is broadly conserved across metazoans.
Response: We thank Reviewer 3 for their thoughtful and constructive comments. In response to the suggestion to expand the evolutionary comparison, we have included a new supplemental figure (Supplemental Fig. S1) showing that PARG proteins from Danio rerio and Caenorhabditis elegans also contain intrinsically disordered tails, supporting the idea that this feature is conserved across metazoans. We have also revised the corresponding section of the Results to incorporate this broader evolutionary perspective.
R3: While the authors emphasize the divergence between human and Drosophila PARG tail sequences, a direct sequence comparison is not shown. It would be useful to identify whether short functional motifs (e.g., nuclear localization signals, SUMOylation sites) are present in both species despite low overall similarity. A functional domain swapping experiment,replacing the Drosophila tail with the human counterpart and testing for rescue of nuclear localization or chromatin association, could provide stronger evidence for conserved functionality.
Response: We thank Reviewer 3 for this comment. We previously published the presence of motif on the tails alongside with the alignment between mammalian and Drosophila PARG (see PMID: 34949666 or reference 6 in the manuscript). None of the identified motifs are conserved as shown in the new Supplemental figure 2 that exhibit the sequence alignment between Human, Drosophila, C.elegans, and D.rerio PARG disordered tails. We modified the result section to clarify this statement.
We agree that a domain-swapping experiment between the Drosophila and human PARG disordered tails could provide interesting insight into potential functional conservation despite sequence divergence. In the present study, our focus was to establish the role of the native Drosophila disordered tail in regulating localization, chromatin association, and enzymatic activity, and to link these functions to distinct classes of interacting proteins. Building on this framework, domain-swapping approaches will be well suited for future studies aimed at directly testing the degree to which tail function is conserved despite sequence divergence.
R3: The proposed role of the tail in regulating nuclear import and export is plausible based on the MS data and subcellular localization patterns. Mutagenesis of predicted importin-binding motifs (e.g., PY-NLS-like elements) and testing their effect on nuclear localization would help determine whether importin recognition is essential.
Response: We thank Reviewer 3 for this constructive suggestion. In the revised manuscript, we have added a new Figure 4 investigating the functional importance of the predicted nuclear localization signal (residues 685–710) within the disordered tail. Specifically, we tested the effects of deleting this sequence (PARG-Δ685–710) and of forcing nuclear import of the catalytic domain using a constitutive NLS (PARG-Cat-cNLS). These experiments demonstrate that the disordered tail controls not only PARG nuclear–cytoplasmic trafficking, but also the accurate subnuclear targeting, which is critical for the full enzymatic activity of PARG. The Results and Discussion sections have been updated to reflect and clarify these new results.
R3: The identification of SUMO pathway components in the interactome is intriguing but not fully explored. Mutation of candidate SUMOylation sites (e.g., K-to-R substitutions) and subsequent analysis of PARG localization or stability would help clarify.
Response: We thank Reviewer 3 for this insightful comment. While direct mutagenesis of the predicted SUMOylation sites was not performed here, our work focuses on uncovering a novel regulatory mechanism of PARG mediated by its intrinsically disordered tail. As shown in Figures 1–5, deletion of this region disrupts localization, impairs chromatin association, reduces catalytic activity, and decreases organismal viability. To further define its regulatory network, we performed mass spectrometry, identifying 289 proteins specifically associated with the catalytic domain and 191 with the disordered tail. Gene ontology analyses of these interactors are aligned perfectly with the phenotypes observed upon deletion of either domain; notably, both domains are essential for precise chromatin binding (Fig. 5). The SUMOylation machinery constitutes only one subset within the broader network of proteins interacting with the disordered tail.
R3: The proposed model in which cytoplasmic Stub1 mediates degradation of PARG is only based on MS data. Including cycloheximide chase assays comparing the degradation kinetics of PARG-WT and PARG-Cat would provide more direct support for this mechanism.
Response: We thank Reviewer 3 for this suggestion. Because our experiments are performed in vivo, cycloheximide chase assays are not feasible, as cycloheximide treatment would cause early embryonic lethality. Moreover, our data already demonstrate that the disordered tail plays roles extending beyond nuclear–cytoplasmic trafficking, including regulation of PARG subnuclear localization and enzymatic activity. We have revised the Discussion section to emphasize these broader functional roles. We appreciate the reviewer’s insightful feedback, which has helped strengthen the manuscript.
Round 2
Reviewer 1 Report
Comments and Suggestions for Authors
The authors' responses to the comments are satisfactory.
Reviewer 2 Report
Comments and Suggestions for Authors
After the revision, I think this paper is good for the publication.
Reviewer 3 Report
Comments and Suggestions for Authors
The author has adequately addressed the comments I raised. I have no further suggestions.